# Automatic classification of mice vocalizations using Machine Learning techniques and Convolutional Neural Networks

**Marika Premoli**[1☯]*, **Daniele Baggi**[2☯], **Marco Bianchetti**[2☯], **Alessandro Gnutti**[2], **Marco Bondaschi**[2], **Andrea Mastinu**[1], **Pierangelo Migliorati**[2], **Alberto Signoroni**[2], **Riccardo Leonardi**[2], **Maurizio Memo**[1], **Sara Anna Bonini**[1]

**1** Department of Molecular and Translational Medicine, University of Brescia, Brescia, Italy, **2** Department of Information Engineering, University of Brescia, Brescia, Italy

☯ These authors contributed equally to this work.
* m.premoli002@unibs.it

**Data Availability Statement:** Datasets used in this paper are available at the following link: https://osf.io/ukgcm/.

## Abstract

Ultrasonic vocalizations (USVs) analysis is a well-recognized tool to investigate animal communication. It can be used for behavioral phenotyping of murine models of different disorders. The USVs are usually recorded with a microphone sensitive to ultrasound frequencies and they are analyzed by specific software. Different calls typologies exist, and each ultrasonic call can be manually classified, but the qualitative analysis is highly time-consuming. Considering this framework, in this work we proposed and evaluated a set of supervised learning methods for automatic USVs classification. This could represent a sustainable procedure to deeply analyze the ultrasonic communication, other than a standardized analysis. We used manually built datasets obtained by segmenting the USVs audio tracks analyzed with the Avisoft software, and then by labelling each of them into 10 representative classes. For the automatic classification task, we designed a Convolutional Neural Network that was trained receiving as input the spectrogram images associated to the segmented audio files. In addition, we also tested some other supervised learning algorithms, such as Support Vector Machine, Random Forest and Multilayer Perceptrons, exploiting informative numerical features extracted from the spectrograms. The performance showed how considering the whole time/frequency information of the spectrogram leads to significantly higher performance than considering a subset of numerical features. In the authors' opinion, the experimental results may represent a valuable benchmark for future work in this research field.

## Introduction

Rodent models are good tools for scientific research because they allow to study and understand the biological mechanisms underlying the different pathologies. Behavioral alterations in animal models offer markers for the symptoms of the human diseases [1]. The study of mice ultrasonic communication has become an important instrument for behavioral phenotyping in several complex pathologies, such as autism spectrum disorders [2, 3]. Communication in

**Funding:** This work was supported by the research grant from the University of Brescia.

**Competing interests:** The authors have declared that no competing interests exist.

laboratory mice mainly occurs through the emission of ultrasonic vocalizations (USVs) with a frequency above the range of human hearing (20 kHz) [4]. USVs are produced during development and at adulthood in different social contexts: pups maternal separation [5, 6], juveniles play [7], adults mating and social investigation [8, 9]. USVs represent a communicative signal and they are an index of social interest and motivation [10]. Communication alterations are found in murine models of different diseases, in particular models of neurodevelopmental disorders (NDDs) and autism spectrum disorders (ASD) [11–13]. Despite the growing attention given to USVs as a tool to investigate behaviors in these mouse models, little is known about the specific meaning of different types of ultrasonic calls.

The USVs are generally recorded with an ultrasound sensitive microphone and they are analyzed by specific software applications. Each syllable can be classified manually based on specific features, such as frequency, duration, amplitude and general shape. The manual classification provides a detailed characterization, but it is very time-consuming and may be subject to personal interpretation. Even using specific and professional tools, this task usually takes a lot of time and it appears to be only partially automatized. Indeed, it is possible to automatically analyze quantitative parameters (i.e., number, duration, peak amplitude and peak frequency of calls), but it is more difficult to evaluate qualitative parameters, such as the different typologies of ultrasonic calls. For these reasons, it could be very interesting to find a method that automatically processes vocalizations starting from the audio tracks. This fundamental and challenging step could speed up the analysis of ultrasonic communication and also provide insight into the meaning of different USVs.

Some automated systems exist and they can be used to analyze rodent USVs, such as MUPET system for mice [14], WAAVES system for rats [15], and the more recent DeepSqueak for several animals calls [16]. All these programs are able to automatically segment audio files into calls and background noise. After the segmentation, they then apply a classification algorithm in order to label the retrieved vocalizations. In particular, WAAVES performs a binary classification with two categories, namely Frequency modulated calls and Flat calls. Conversely, MUPET and DeepSqueak perform an unsupervised clustering with a higher number of categories. Regarding the number of categories, MUPET can only cluster into a known a priori number of classes that must be set by the user, while DeepSqueak can automatically detect the best-fitting number of clusters using statistical methods. Moreover, MUPET cannot apply its models to datasets that are different than the ones for which the models are originally designed for, therefore different models are needed for different datasets. On the contrary, our work aims to develop an *ad hoc* method for automatic USVs classification on the basis of the well-known USVs classification pattern published by Scattoni and colleagues [17]. Therefore, the main difference with the other methods is that our classification is performed on a predefined set of call types. Moreover, we started our classification assuming an audio track already segmented in vocalizations / non vocalizations. It can be stated that the other approaches follow a bottom-up paradigm, whereas our method applies a top-down approach [18], as also employed in the very recent work described in [19], in which the authors propose a deep learning method to classify the sex and the strain from mouse USVs.

It is worthy to note that recent works suggest that the USVs categorization into a discrete number of classes may be a too arduous task, given a not always clear separation of the classes. For example, Sainburg and colleagues [20] report that USV vocal elements can be either less clearly stereotyped or less temporally distinct, increasing the difficulty in identifying the correct discrete class. However, manual classification can be performed within a more than reasonable margin of error, especially when the general shape of the spectrogram is analyzed. Indeed, as we will point out in the discussion of the results, the performance of the proposed methods shows a good ability for USVs categorization, indicating that well defined models can

actually lead to significant results also for discrete vocalizations classification task. Interestingly, also Vogel and colleagues investigated some classification methods aimed to categorize USV calls into a discrete number of classes [21]. In particular, the authors proposed four feature selection algorithms to select optimal feature subsets, which are then fed into Support Vector Machines (SVM) and Random Forests (RF). However, there are two main differences between the work described in [21] and the one proposed here. First, we conducted our experiments on a dataset composed by 48699 samples instead of 225 samples as done in [21]. The sample size is a crucial consideration for quality research, and generally, larger dataset cardinality allows to better determine the average values of data and reduce potential errors from testing a small number of possible outliers. Furthermore, in our work we also explored the classification ability of Artificial Neural Networks, extending the analysis of traditional machine learning algorithms, such as SVM and RF, as done in Vogel's paper. To our knowledge, our work is the first to propose a deep learning architecture aimed to classify the ten categories originally introduced in [17].

For automatic classification task, we customized and used two deep learning methods, namely, a Convolutional Neural Network (CNN) and a Multilayer Perceptron (MP), in addition to two machine learning techniques, i.e., Support Vector Machines (SVM) and Random Forests (RF). The dataset was manually built using the Avisoft software that offers some tools to visualize and manually segment the audio track in the form of spectrograms. The retrieved USVs were then processed in order to extract some informative features needed to train the learning methods. Experimental results appear to be encouraging, even if future work may further improve the overall performance.

## Materials and methods

### Animals

NF-kB p50 knock-out (KO) mice (B6;129P2-Nfkb 1tm 1 Bal/J) and control wild-type (WT) mice (B6;129PF2) were purchased from The Jackson Laboratories (Bar Harbor, ME, USA). KO mice have a deletion of NFkB1 gene, coding for the NF-kB p50 subunit and they are a mouse model of neurodevelopmental disorders, as we have previously demonstrated [22, 23]. Animals were maintained in standard cages in a 12 hours light/dark cycle (light phase: from 8:00 a.m. to 8:00 p.m.) with ad libitum access to food and water. p50 KO and wild-type (WT) pups were used for experiments; they were obtained in our animal facility from mating mice. Behavioral procedures were performed in conformity with the European Communities Council Directive of 1986 (86/609/EEC), and approved by the Italian Ministry of Health, Animal care and use Committee of the University of Brescia.

### USVs recording and analysis

Ultrasonic vocalizations of 40 WT and 39 KO pups at postnatal day (PND) 6, 8 and 12 were recorded after a maternal separation of 3 minutes, as previously described [11, 24, 25]. An ultrasound microphone (CM16/CMPA ultrasound microphone, Avisoft Bioacoustics, Berlin, Germany) sensitive to frequencies of 10–180 kHz and suspended 20 cm above the container that hold the pups, was used. The microphone was connected through an external audio interface (USG 116, Avisoft Bioacoustics, Berlin, Germany) to a personal computer.

Vocalizations were recorded at 250 kHz in 16-bit format with Avisoft Recorder (version 4.2; Avisoft Bioacoustics, Berlin, Germany) and analyzed by Avisoft SasLabPro (version 5.2; Avisoft Bioacoustics, Berlin, Germany) after a fast Fourier transformation (512 FFT-length, 100% frame, Hamming window and 75%-time window overlap). Spectrograms were obtained with a frequency resolution of 488 Hz and time resolution of 0.512 milliseconds (ms). A high

pass filter of 30 kHz was employed to delete background noise. USVs detection was performed with an automatic threshold-based algorithm (threshold: -80 dB) and a hold time mechanism (hold time: 10 ms). Number, duration, peak frequency and peak amplitude were calculated for each vocalization [26]. Finally, spectrograms of each pup were examined manually by the operator and USVs were categorized into 10 different classes including complex, harmonics, two syllable, upward, downward, chevron, short, composite, frequency steps and flat (Fig 1).

## Mice vocalizations dataset

The dataset was built manually by the human operator using the Avisoft software segmenting the spectrograms computed by the program on both the time and frequency axis. On the basis of the user's segmentations, the software saves a table in whose rows are reported each identified segment with some features related to that. This work was performed on the audio tracks of WT and KO pups.

Each USV segment was then assigned to a specific class, based on the characteristics that define the different USVs type [17]. The complete dataset is composed by both WT and KO pups' vocalizations for a total amount of 48699 labelled segments.

## Description of the experiments

The vocalizations classification task was to assign a suitable label to each vocalization, where the label identifies the class of the vocalization. In this analysis the possible labels were ten, referred to the different USVs categories identified by Scattoni [17]. The classification of a vocalization was performed using some specific features that properly characterize the vocalization as belonging to a certain class. Therefore, the first step consisted of the so-called features extraction, followed by the actual classification step. For this purpose, several methods were used, such as CNN, MP, SVM and RF. In order to assess the investigated classification algorithms, 10-fold cross-validation (CV) has been applied, ensuring the statistical soundness to the results. Dataset was split into training/validation and test datasets in according to 80%-20% proportion.

The classification analysis was performed using both Matlab (R2019b) and Python (version 3.7) programming languages. We used the Statistics and Machine Learning Toolbox and the Deep Learning Toolbox, when working in Matlab environment, while for Python we exploited some popular libraries such as: scikit-learn [27], which is a machine learning library that provides various classification, regression and clustering algorithms (including Support Vector Machines and Random Forests), Keras [28], which is an open source high-level neural network library, and TensorFlow [29], which is an open source machine learning framework for high performance numerical computation.

In the following, we will illustrate the details of the experiments carried out for this task, focusing the attention on features extraction, data pre-processing and the definition of the investigated classification methods.

**Features extraction.** From an initial set of measured data, some values called features are extracted. They are informative, non-redundant and contain the relevant information of the input data. In this work, two types of feature sets were used:

• *Spectrograms*, consisting of 192x834 pixels images that represent a joined 2-D visual representation of the frequency and time information of a data signal. In particular, a spectrogram is the visual representation of the signal frequencies that varies with time. The two axes of the corresponding image thus represent time and frequency. The amplitude of a specific frequency at a certain time is identified by the brightness of the corresponding pixel in the image.

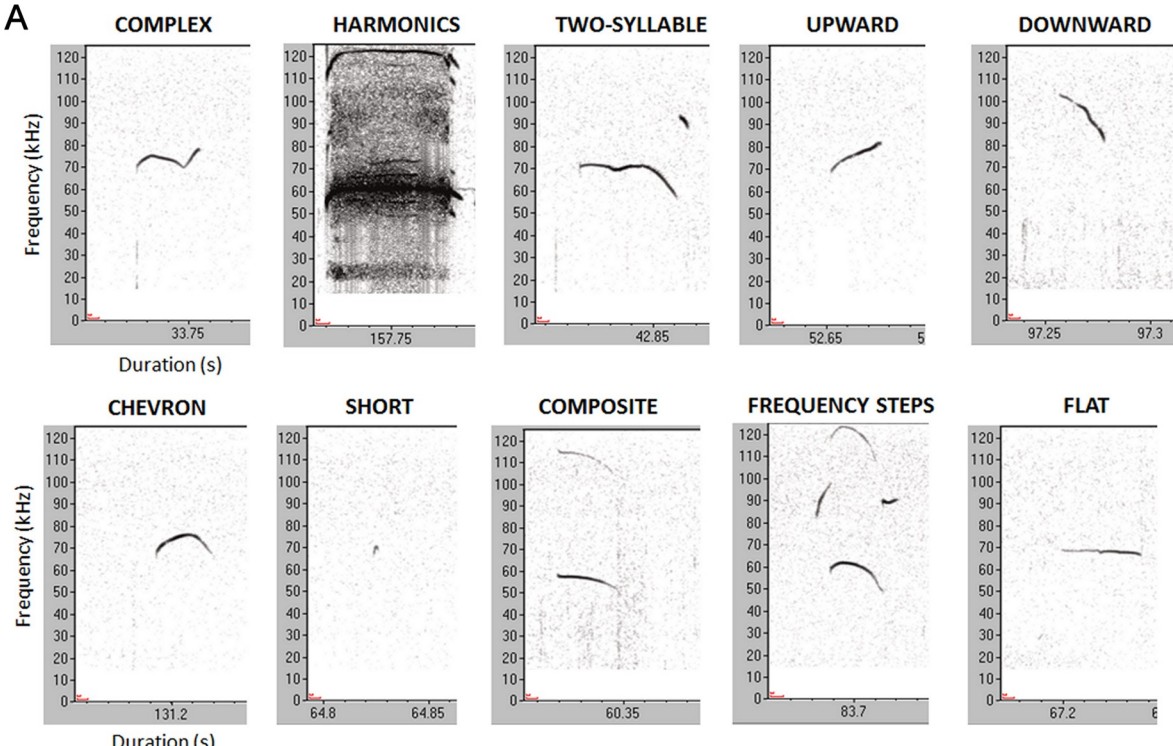

**Fig 1. Examples of USVs categories emitted by mice.** (A) Representation of typical spectrograms of different USVs categories. (B) Description of the features of the 10 USVs types.

- So called "*Standard features*", that are 20 features obtained by the Avisoft software organized in a 20-dimensional vector. They are: duration, peak to peak, quart 50 (start), quart 75 (start), peak frequency (start), peak amplitude (start), peak frequency (end), peak amplitude

(end), peak frequency (max), peak amplitude (max), peak frequency (mean), peak amplitude (mean), peak frequency (min-entire), peak frequency (max-entire), peak amplitude (max-entire), peak frequency (mean-entire), peak amplitude (mean-entire), peak freq (standard deviation-entire), peak amplitude (standard deviation-entire) and max freq (standard deviation-entire). Specifically, the main features are the vocalization duration, measured in milliseconds, the difference between the maximum and minimum values of the audio file, the quartiles (50% and 75%) of the spectra, and the peak frequency and peak amplitude, both taken in different configuration related to the "min", "max", and "mean" values obtained in different ways. In particular, "start" and "end" refer to the value of the parameter in the actual first and last temporal position of the spectrogram, while "entire" refers to the parameter value computed over the whole spectrogram, which means for all spectra between the start and the end of a spectrogram. So, for instance, the peak frequency (max entire) corresponds to the highest peak frequency that occurred between the start and end of a spectrogram vocalization. In order to distribute each individual feature as a Gaussian with zero mean and unit variance, feature standardization was applied. This precaution is a common requirement for many machine learning estimators, since it avoids that a feature with a high variance may cause the estimator to be unable to learn from other features.

**Class imbalance handling.** A critical problem for learning algorithms is represented by class imbalance. Depending from the entity of the asymmetry of the classes cardinality, the classification process may result more or less skewed. Considering our dataset, it is straightforward to note that the number of recorded vocalizations differs class by class, since mice emit different vocalizations (both quantitatively and qualitatively) based on the environmental context. As a direct consequence of this fact, the final dataset is indeed imbalanced. In particular, the minority class is represented by the *upward* category, which consists of just 1199 samples, while the majority class, i.e., the *frequency steps* call, contains 12136 samples. The ratio between these two classes' size suggests a potential risk of losing the ability to model the minority class.

Several strategies can be used to handle this problem: in our work, we tested two approaches for the CNNs, which use the spectrograms as input data, and two different approaches for the rest of the investigated algorithms, which take numerical features as input.

For the CNN design, data augmentation and loss function weighting were analyzed, respectively. In data augmentation, an offline generation of new images was applied in order to balance the source dataset, namely by generating new examples just for the under-represented classes. Note that this operation also leads to an enrichment of the diversity of training samples, and so it may help to reduce overfitting; however, overfitting prevention will be discussed next in the paper. Alternatively, a properly weighted loss function was considered in the training phase, in order to give a specific weight to each class based on the corresponding cardinality: in particular, the smaller is the size of a class, the larger is the weight assigned to its samples. In the end, the experimental results showed a higher ability to fairly represent all the classes when the second strategy was applied, and so data augmentation was discarded, and class weighting was used for training the best model.

Moving the focus on the algorithms that receive numerical values as input features, random oversampling and down-sampling were tested, respectively. In random oversampling, some training examples were randomly duplicated from the minority classes and then added to the training dataset, while the down-sampling technique consists of a random selection of examples from the majority class and the consequent deletion of them from the training dataset. When adopting this second approach, worse performance was obtained with respect to the one provided from the oversampling method, that was indeed used for training the final models. The bad results obtained when downsampling the training set show that the loss of such data made it harder for the network to learn the decision boundaries between the classes.

**Proposed classification methods.** *Multilayer perceptrons*. MP are learning algorithms composed by a set of connected nodes called artificial neurons or perceptrons [30]. The nodes in a MP are grouped into layers, namely, a single input layer followed by one or more hidden layers, and an output layer. Each layer is connected to the adjacent ones by a set of connections (called *edges*) between their nodes. A non-linear function is assigned to each neuron and a weight is associated to each edge. Each artificial neuron processes its input signal and then it transmits the resulting information to the following artificial neurons connected to it. The MP aims to solve a task, without being programmed with any task-specific rules. This is possible thanks to a learning phase in which many training examples are processed by the network. During the learning phase each training example, for which the expected network output is known, flows into the network and the edges weights are modified in order to obtain an output as close as possible to the expected one.

*Convolutional neural network*. CNN is a deep learning algorithm which takes an image as input, and by capturing its various properties, it is in the end able to distinguish a given image from other ones. In essence, it can successfully identify spatial dependencies through the application of relevant filters. A CNN is basically composed of three types of layers. The first is the convolutional layer, that is composed by a series of filters (or kernels) involved in carrying out the convolution operation. Different filter and stride sizes can be properly tuned for each convolutional layer, in order to extract significant high-level features. The second kind is the pooling layer, useful for decreasing computational times and extracting dominant features. Depending on the employment of max or average pooling, the maximum or the average of all the values from the space covered by the kernel is returned. The convolutional and the pooling layers taken together typically refer to the i-th layer of the CNN, and the architecture can be formed by one or more consecutive layers, defining the network topology. Finally, the model ends with one or more of the third type of layer, i.e., the fully-connected layer (corresponding to one layer of the MP model), that may help to detect insightful non-linear combinations of the features as computed by the preceding convolutional layers and that are therefore responsible for the final classification of the input image.

*Support vector machines*. SVMs are machine learning algorithms that fall in the supervised learning category. They can be applied to solve both classification and regression problems [31]. At first approximation, given a set of binary training samples, SVM is an algorithm that aims to split the data into two classes, by finding the line that separates those classes samples. In this case the SVM model becomes a non-probabilistic binary classifier. The samples of a SVM model can be represented as points in space, mapped so that, in the best-case scenario, the samples of the different categories are divided by a clear large gap. New samples are then mapped into the same space and assigned to a category based on the position of the correspondent points with respect to the gap between the two classes [31]. In order to carry out multiclass classification, ensembles of SVM classifiers were used. Commonly, the choice is done between two ensemble architectures:

• One Versus All (OVA) architecture, in which every classifier is responsible for the identification of one of the $N$ classes, and so $N$ SVM classifiers are needed. Thus, for our task, 10 SVM classifiers were trained. Maximum confidence strategy was used, so the output class was taken from the classifier with the largest positive confidence.

• One Versus One (OVO) architecture, in which every classifier is responsible for the choice between two of the N classes, and so $N(N-1)/2$ SVM classifiers are needed. Thus, for our task, 45 SVM classifiers were trained. The final classification is carried out by majority voting.

In addition to performing linear classification, SVM can efficiently perform also a non-linear classification using the so-called *kernel trick* [32], by implicitly mapping its inputs into high-dimensional feature spaces.

*Random forests.* RF is an ensemble learning method for classification and other tasks that operates by constructing a multitude of decision trees during the training phase. Its output is computed by taking into account the outputs of all the trees in the forest [33]. Therefore, this learning method uses a decision tree as a predictive model which aims to go from the observations about an item (represented in the tree branch) up to the conclusions about the item's target value (represented in the tree leaf). In the case of the classification task, the trees are called classification trees and the target assumes discrete values, called classes. Finally, in the classification tree structures, leaves represent the class labels, and branches represent conjunctions of features that lead to those class labels.

## Results and discussion

The classification algorithms previously described were tested and the performance are discussed in this section. The results are reported in terms of accuracy, precision, recall and confusion matrix. The accuracy is computed as the number of right guesses (that represent the elements of the main diagonal of the confusion matrix) divided by all the predictions. The precision is computed as the number of true positives divided by the sum of the true and false positives (columns of the confusion matrix), while the recall is computed as the true positives divided by the sum of the true positives and false negatives (rows of the confusion matrix).

Before reporting the results, we describe in detail the procedure to obtain the best-performing setting for the examined classification techniques.

### Classification via multilayer perceptrons

Several Fully Connected NN were tested with different numbers of neurons and layers. In particular, the following different combination of hidden layers number of neurons were tested: (32, 64), (32, 64, 128), (32, 64, 128, 256), (32, 64, 128,128, 64, 32), (16, 32, 128, 128, 32). All the configurations have an input layer of size 20, and an output layer of size 10. Every layer adopted a Hyperbolic Tangent activation function, but the last one that was implemented with Softmax activation function. Those networks were trained with a balanced dataset composed of the "standard features", using a learning rate of $10^{-4}$, for a number of epochs equal to 500, feeding the whole training set for each epoch. The best performing configuration resulted to be the one composed of 5 fully connected layers, respectively with 20, 32, 64, 128 and 10 neurons.

We also tested two advanced techniques, namely Stacking Neural Network and Auto-Context Neural Network. In the first, the Neural Network (NN) is composed of a small NN called "base learner" whose outputs are used as input of an another network called "meta learner", which goal is to weight the classification of each base learner together with the ground truth of the original input data in order to evaluate the prediction of each base learner, and so to give a more reliable prediction. The base learners were the 5 MP previously discussed. The outputs of those networks were used as input of the meta learner composed by 3 fully connected hidden layers with 1024, 128, and 32 neurons, respectively.

The Auto-Context NN is a network ensemble as the stacking NN previously explained, but in this case also the original sample itself was used as input data of the meta learner, together with the output of the base learners. The test was performed with just only 3 base learners (corresponding to the second, third, and fourth fully connected tested networks) and a meta learner composed by 2 hidden layers with 128 neurons each.

### Classification via CNN

Some custom Convolutional Neural Networks (CNN) were tested with different number of convolutional layers (1–6), fully connected layers (1–3), and filters per layer (4–256).

The best performing model resulted to be composed by 5 convolutional layers constituted by 32, 64, 64, 128 and 128 filters, respectively. The associated kernel sizes were 7x7 and 5x5 in the first two layers, and 3x3 in the successive three. The stride was set to 1 for each layer.

A max-pooling layer was inserted after each of them. The pooling size was fixed to 3x3, while the stride was set to 2x2. The tensor was then flattened, and two fully connected layers were inserted, the first of size 1024 with ReLU activation function, and the second of size 10 with Softmax activation function. We mention that the Adam optimization algorithm has been used as stochastic gradient descent optimizer, and that the uniform distribution has been adopted to initialize the network weights.

This CNN was trained using the spectrograms extracted from each dataset sample as input data. An hyperparameters optimization/tuning was performed in order to find the best setting of parameters such as the number of epochs, the learning rates and the batch size, to mention the most important ones. At the end, training for 200 epochs with learning rate and batch size equal to 0.001 and 32, respectively, led to the best performance.

In addition, some strategies to prevent overfitting were applied, such as the introduction of dropout and batch normalization layers. Dropout deletes a random sample of the activations (namely, makes them zero) during training, while batch normalization helps to coordinate the update of multiple layers in the model. In the final configuration, we inserted a dropout layer after each convolutional/max-pooling layer, except the first one, setting the rates (i.e., the fraction of input units to drop) to 0.2, 0.3, 0.4 and 0.5. Then, a further dropout layer was added after the first fully connected layer (rate equals to 0.5), and finally batch normalization was included before the Softmax activation function of the last fully connected layer. Actually, also preprocess data augmentation was tested for overfitting prevention, by generating new images for the training dataset obtained by random shifts and crops. However, the experimental results showed that data augmentation does not influence significantly the performance. Figs 2 and 3 show the scheme of the final CNN architecture and the table that details the layers composing it, respectively.

## Classification via support vector machines

The SVM classification method was tested with both the OVA and OVO multiclass classification configurations. In this analysis the dataset is composed by the 20-dimensional standard features. As first step a grid search was performed with a 5-fold cross-validation in order to

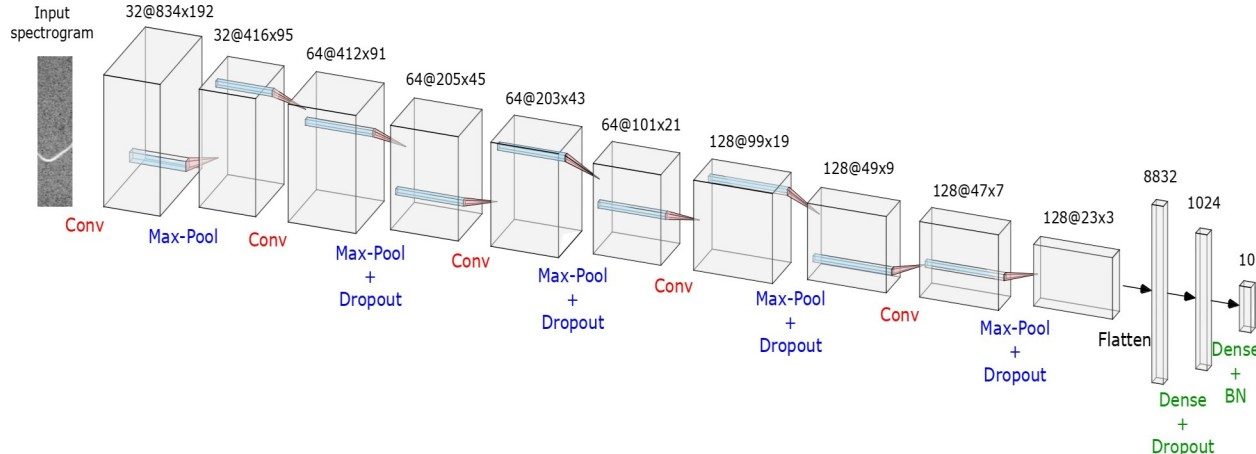

**Fig 2. Scheme of the proposed CNN architecture.** The figure shows the visual overview of the designed CNN architecture.

| Layer | Output shape |
|---|---|
| Convolutional | (192,834,32) |
| Max Pooling | (95,416,32) |
| Convolutional | (91,412,64) |
| Max Pooling | (45,205,64) |
| Dropout | (45,205,64) |
| Convolutional | (43,203,64) |
| Max Pooling | (21,101,64) |
| Dropout | (21,101,64) |
| Convolutional | (19,99,128) |
| Max Pooling | (9,49,128) |
| Dropout | (9,49,128) |
| Convolutional | (7,47,128) |
| Max Pooling | (3,23,128) |
| Dropout | (3,23,128) |
| Flatten | 8832 |
| Dense | 1024 |
| Activation | 1024 |
| Dropout | 1024 |
| Dense | 10 |
| Batch normalization | 10 |
| Activation | 10 |

**Fig 3. Scheme of the proposed CNN architecture.** The figure reports the table that evidences the layers composing the designed CNN architecture; here the output shape is in the form (*width*, *height*, *depth*).

find the best parameters such as the kernel type (radial basis function, linear, polynomial, sigmoid), the features normalization and scaling, the number of features (15, 13, 11, 9, 7, 5, and 3, obtained by applying a Principal Component Analysis) and other parameter values such as the

Gamma (0.0001, 0.0005, 0.001, 0.005, 0.01, 0.1) and C coefficients (1, 10, 50, 100, 1000, 5000, 10000, 100000), which are regular SVM parameters [31]. The set of data used to perform this task is the training set only, corresponding to the 80% of the whole available data.

The best results related to the OVA configuration were obtained using a Radial Basis Function (RBF) kernel, C equal to 5000, Gamma with value of 0.0005, taking into account the normalized feature of the entire input feature space, because when SVM was trained on the features provided by the PCA algorithm lower performance was achieved. The best results related to the OVO configuration were obtained using the same configurations as in the OVA case, with the only difference that a Gaussian kernel was used.

## Classification via random forests

First, we mention that the cross-validation approach during the grid-search of the best parameters was applied only in the SVM classifier. Furthermore, the "standard features" used for RF correspond to the whole 20 original features, and no PCA technique was applied.

Regarding Random Forests method, 200 and 500 trees were trained with the 20-dimensional standard features of the balanced dataset; the best results were obtained with the large forest.

## Performance analysis

The experimental results have clearly shown that when using the whole time/frequency information of the spectrogram, the USVs classification performs significantly better. As a matter of fact, the CNN architecture has outperformed the standard features-based methods, showing a significant ability in distinguishing the USVs calls in the considered classes. For the sake of conciseness, we focus the discussion on the performance analysis of the most-performing classification model, i.e., the CNN, as just mentioned. However, also an overview of the results achieved with the standard features-based techniques will be properly reported.

We divided our experiments in three runs: in the first, the classification was tested on the entire mice vocalizations dataset, while in the second and third ones, we performed the evaluation by considering the USVs produced by KO mice and WT mice, separately. This way, we tried to capture potential correlation between mice belonging to the same strain. For each experiment, 10-fold CV has been applied: Table 1 reports the results in terms of precision, recall and accuracy, indicated as mean ± standard deviation.

Comparing the different experimental runs, it is possible to observe that the results related to the entire dataset and the ones referred to the KO mice dataset appear to be similar, in particular all the considered metrics return more than 78% of performance. Instead, the classification corresponding to the WT mice dataset seems to be slightly more challenging. This fact suggests that USVs derived by WT mice could be more difficult to categorize than the ones derived by KO mice.

Fig 4 shows the corresponding confusion matrices, where the predictions are arranged in ten columns, while in rows are reported the correct classes (*ground truth*). On the main

**Table 1. Test set classification results of CNN architecture considering the entire dataset and the two genotype-based datasets, evaluated in terms of precision, recall and accuracy values.**

|  | Precision (%) | Recall (%) | Accuracy (%) |
|---|---|---|---|
| Entire dataset | 78.6 ± 1.8 | 79.0 ± 1.7 | 78.8 ± 1.7 |
| KO mice dataset | 78.2 ± 1.4 | 78.3 ± 1.5 | 78.3 ± 1.4 |
| WT mice dataset | 75.1 ± 1.5 | 74.8 ± 1.6 | 74.9 ± 1.6 |

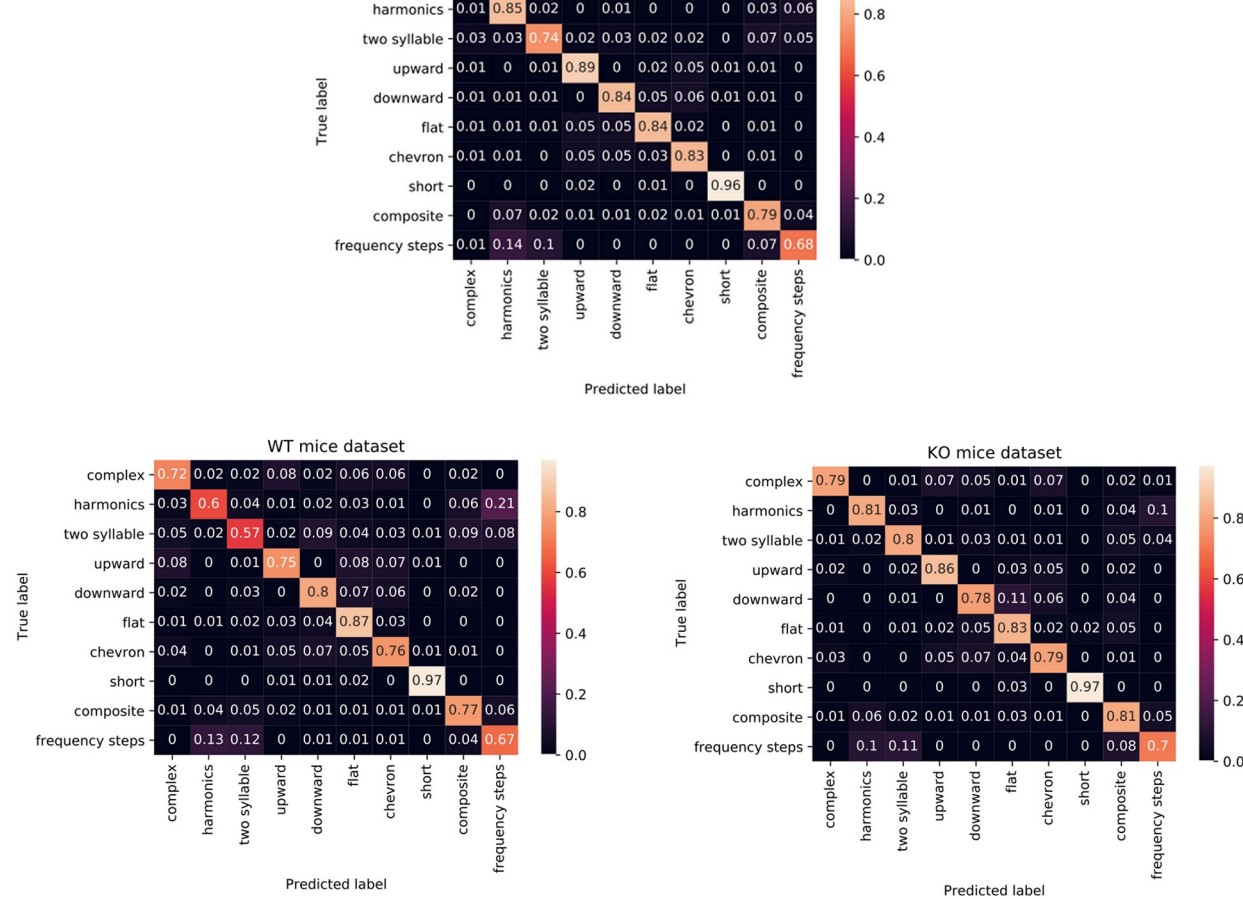

**Fig 4. Confusion matrices related to the classification results obtained testing the proposed CNN for the three datasets.** On both axes there are the vocalization classes; in particular, the x axis refers to the predicted labels, while the y axis refers to the true ones. Notice that each row value is normalized with respect to the number of vocalizations of that row's class. The values in the range (0;1) can be interpreted as percentage values.

diagonal there are the correct guesses, while the off-diagonal elements are all the false positives of each class (moving along the columns) and all false negatives (moving along the rows). The values in the matrices are normalized over the total number of samples belonging to each class (the sum of the values along the row).

Some general considerations can be deduced by analyzing the confusion matrices associated to the three experiments:

• In general, the *short* class seems to be the simplest one to be classified. In fact, the proposed CNN returns very high true positive percentage value related for all the three datasets, reaching at least 95% of accuracy.

• For the KO mice dataset, no class returns an accuracy value lesser than 0.7, which corresponds to *frequency steps*. Furthermore, all the other classes achieve a correctness always greater than 0.78.

• For the WT mice dataset, the classes responsible for the slightly lesser performance are *harmonics*, especially confused with *frequency step*, and *two syllable*.

As mentioned before, we also recap the results associated to the rest of the classification methods studied in our work (see Table 2); to streamline the paper, we just report the

**Table 2. Summarization of the test set classification results for the standard features-based algorithms considering the entire dataset, evaluated in terms of precision, recall and accuracy values.**

| Method | Precision (%) | Recall (%) | Accuracy (%) |
|---|---|---|---|
| SVM OVA | 70.0 ± 0.7 | 72.1 ± 0.6 | 73.9 ± 0.8 |
| SVM OVO | 64.9 ± 0.5 | 68.9 ± 0.4 | 64.8 ± 0.5 |
| Random Forest | 66.3 ± 0.5 | 65.0 ± 0.6 | 65.6 ± 0.6 |
| ANN Fully Connected | 65.8 ± 0.3 | 66.3 ± 0.4 | 65.5 ± 0.3 |
| ANN Stacking | 69.1 ± 0.4 | 69.5 ± 0.4 | 68.4 ± 0.4 |
| ANN Auto-Context | 68.5 ± 0.8 | 68.9 ± 0.7 | 67.8 ± 0.7 |

performance related to the entire mice vocalizations dataset, since the strain-based performances are similar to the ones obtained for the entire dataset, and so they do not add significant extra information.

As a final note, in order to provide a comparison with the recent work described in [21], we repeated the experiments by removing the class *harmonics*, as done in Vogel's paper. The results obtained for this 9-classes classification are reported in Table 3, in which the precision, recall and accuracy values regarding the CNN model are indicated, and in Fig 5, where the corresponding confusion matrices are illustrated. The accuracy significantly increases (about 4% for all the three datasets), reaching a performance that arises to be comparable with respect to the Vogel's paper [21]. However, we want to point out that given the notable size of our dataset, our results benefit from a highly valuable statistical significance and may represent a valuable benchmark for future work in this research field.

## Conclusions

In this paper, we have addressed the problem of USVs classification. Specifically, we have proposed and evaluated various methods for an automatic supervised classification, based on the well-known classification pattern proposed by Scattoni and colleagues [17]. The reference dataset used in this work has been manually defined using the Avisoft software, that offers some tools useful to visualize and manually segment the spectrograms of the considered audio tracks. Then, the segmented USVs have been elaborated in order to extract some informative features necessary to train the supervised machine learning methods. For the classification task, a Convolutional Neural Network architecture has been trained receiving as input the spectrogram images associated to the segmented audio files. Furthermore, also a number of numerical feature-based learning algorithms have been evaluated, such as Multilayer Perceptrons, Support Vector Machine and Random Forest.The performance showed that the exploiting of the whole time/frequency information of the spectrogram leads to significantly higher performance than considering only a subset of numerical features. The results appear to be very encouraging, and we believe that they may represent a valuable benchmark for future work in this research field. Nonetheless, further significant improvements could be obtained by increasing the dimensions of the dataset and working on the selection of the spectrogram's

**Table 3. Test set classification results of CNN architecture considering the entire dataset and the two genotype-based datasets, evaluated in terms of precision, recall and accuracy values using 9 classes of calls.**

| | Precision (%) | Recall (%) | Accuracy (%) |
|---|---|---|---|
| Entire dataset | 82.8 ± 1.7 | 82.9 ± 1.8 | 83.1 ± 1.6 |
| KO mice dataset | 82.8 ± 1.5 | 82.6 ± 1.6 | 82.7 ± 1.6 |
| WT mice dataset | 79.4 ± 1.8 | 79.5 ± 1.7 | 79.5 ± 1.9 |

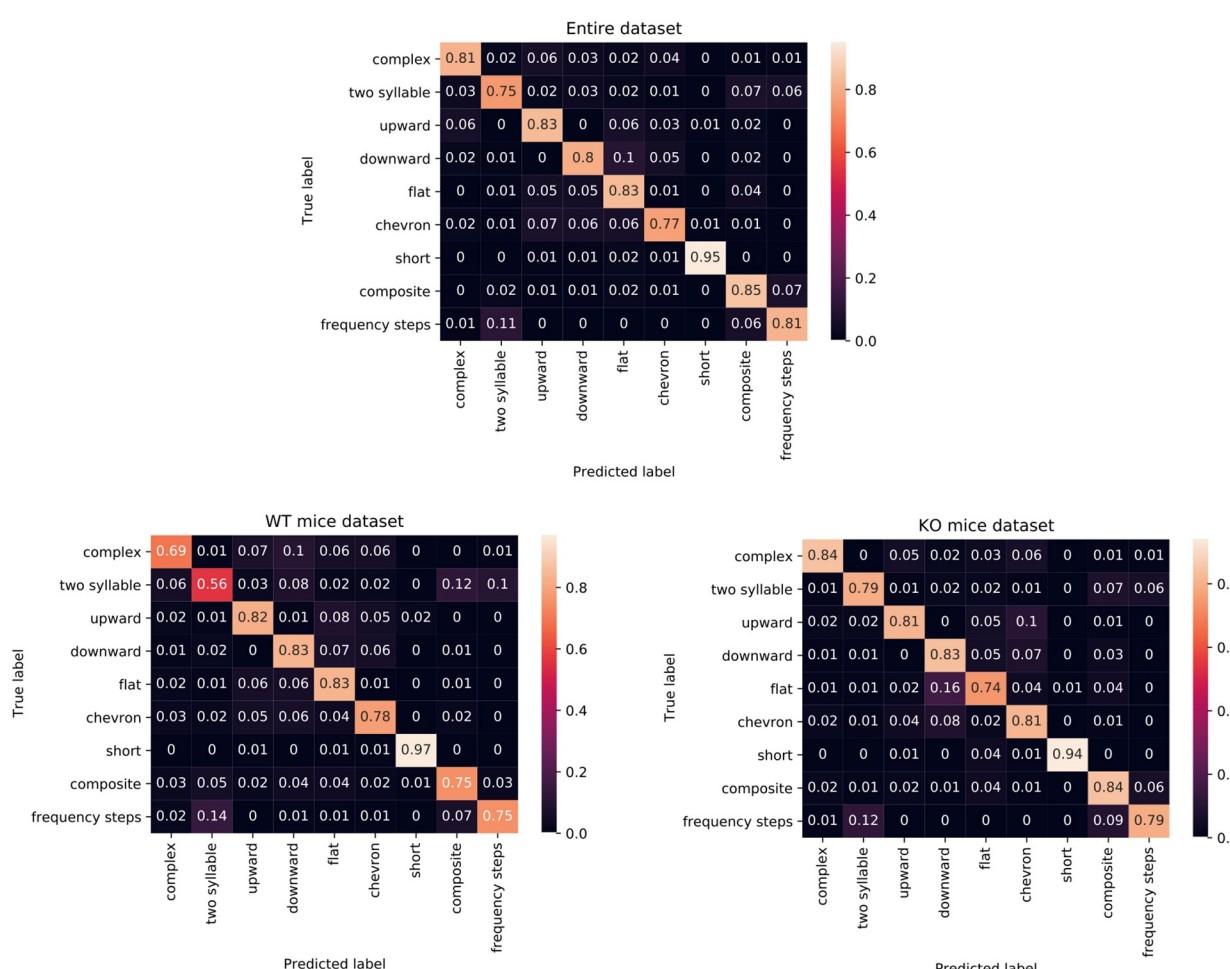

**Fig 5. Confusion matrices related to the 9-categories classification results obtained testing the proposed CNN for the three datasets.** On both axes there are the vocalization classes; in particular, the x axis refers to the predicted labels, while the y axis refers to the true ones. Notice that each row value is normalized with respect to the number of vocalizations of that row's class. The values in the range (0;1) can be interpreted as percentage values.

features. Current research is also devoted to the definition of a completely automatic system which automatically performs also the segmentation task. The final set up of an automatic classification method will definitely solve the current main problems of USVs manual classification: its being a time consuming process and operator bias.

## Author Contributions

**Conceptualization:** Marika Premoli, Daniele Baggi, Marco Bianchetti, Pierangelo Migliorati.

**Funding acquisition:** Pierangelo Migliorati, Riccardo Leonardi, Maurizio Memo.

**Investigation:** Marika Premoli, Daniele Baggi, Marco Bianchetti, Alessandro Gnutti, Marco Bondaschi.

**Writing – original draft:** Marika Premoli, Daniele Baggi, Marco Bianchetti.

**Writing – review & editing:** Marika Premoli, Daniele Baggi, Marco Bianchetti, Alessandro Gnutti, Marco Bondaschi, Andrea Mastinu, Pierangelo Migliorati, Alberto Signoroni, Riccardo Leonardi, Maurizio Memo, Sara Anna Bonini.

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
