## [Decision Letter · Decision Letter 0]

17 Jun 2020

PONE-D-20-09612

Automatic classification of mice vocalizations based on different machine learning methods

PLOS ONE

Dear Dr. Premoli,

Thank you for submitting your manuscript to PLOS ONE. After careful consideration, we feel that it has merit but does not fully meet PLOS ONE’s publication criteria as it currently stands. Therefore, we invite you to submit a revised version of the manuscript that addresses the points raised during the review process.

The authors should seriously revise the manuscript explaining advantages and technical details of the approaches they used and to address the following specific issues.

Although the authors made significant effort in optimizing the hyperparameters of SVM and RF classifiers, the set of features chosen representing the vocalizations seems to be inadequate for the task of spectrogram shape classification. Please, revise introduction, methods, and discussion to address this issue.

The detailed description of the architecture of ANN should be provided. Did you use use any of the overfitting-prevention technique ?

Major

1. For statistical confidence, the cross validation technique(e.g. 10-fold) should be applied for model assessment and the results should be shown together with their confidence intervals

2. The feature set used for RF, SVM and non-convolutional ANN input doesn't seem to be adequate for the desired task. The authors should further justify or extend the feature set and/or use entire frequency envelope data as the input.

3. The architecture of ANNs should employ widely used techniques for overfitting prevention: batch normalization/dropout.

4. Discuss advantages of the approaches based on experimenter derived call categories versus un-biased call classification.

Sangiamo, D.T., Warren, M.R. & Neunuebel, J.P. Ultrasonic signals associated with different types of social behavior of mice. Nat Neurosci 23, 411–422 (2020). https://doi.org/10.1038/s41593-020-0584-z

5. Please, compare your results to Vogel et al. 2019 which achieved 85% recall. What are the advantages of your approaches?

Vogel, A.P., Tsanas, A. & Scattoni, M.L. Quantifying ultrasonic mouse vocalizations using acoustic analysis in a supervised statistical machine learning framework. Sci Rep 9, 8100 (2019). https://doi.org/10.1038/s41598-019-44221-3

6. Please, make a statement concerning sharing the training data, code, and finalized classifiers.

Minor

1. SVM OVA approach requires confidence estimation. Which confidence estimation approach was used?

2. ANN architecture should be described more thoroughly: convolutional layers kernel sizes, strides, dropout/batch normalization usage, stochastic gradient descent optimizer used, weight initialization strategy.

3. Strain data is not used in the analysis anyhow.  It would be interesting to compare the accuracies of two strains' shape prediction results taken separately.

Typos

96 'can not applies'

We look forward to receiving your revised manuscript.

Kind regards,

Gennady Cymbalyuk, Ph.D.

Academic Editor

PLOS ONE

Journal Requirements:

Reviewers' comments:

Reviewer's Responses to Questions

**Comments to the Author**

1. Is the manuscript technically sound, and do the data support the conclusions?

Reviewer #1: Partly

Reviewer #2: Yes

2. Has the statistical analysis been performed appropriately and rigorously? 

Reviewer #1: No

Reviewer #2: Yes

3. Have the authors made all data underlying the findings in their manuscript fully available?

Reviewer #1: No

Reviewer #2: No

4. Is the manuscript presented in an intelligible fashion and written in standard English?

Reviewer #1: Yes

Reviewer #2: Yes

5. Review Comments to the Author

Reviewer #1: The authors developed several classifiers capable of classifying individual vocalizations according to their spectrographic

shapes as described by Scattoni et al, 2008. The work employs a quite comprehensive set of machine learning methods

including Support Vector Machines(SVM), Random Forests(RF), fully connected and convolutional artificial neural networks (ANNs, CNNs).

Two-dimensional spectrograms were used as the input data for CNN-based classifiers. For non-convolutional ANN, SVM and

RF classifiers the vocalizations were represented as the sets of 16 automatically extracted features.

Although the authors made significant effort in optimizing the hyperparameters of SVM and RF classifiers, the set of features chosen

representing the vocalizations seems to be inadequate for the task of spectrogram shape classification.

It can be hard to differentiate 'chevron' from 'complex' using only overall and marginal max/mean/min values. The confusion

matrices provided in the paper confirm that.

The description of the architecture of ANN provided by the authors lacks details. Two convolutional layers only

can be insufficient for good performance in image classification tasks. The authors don't mention whether they use any of

the overfitting-prevention techniques such as batch normalization, dropout, weight regularization and data augmentation.

The fact that the CNN applied to full spectrograms performs worse than SVM applied to the ambiguous feature set

indicates that more effort can be invested in the optimization of CNN architecture and the training protocol.

Note also the performances shown by CNNs in much more complex image classification tasks (CIFAR100) and

the RF performance demonstrated by Vogel at al, 2019 solving a similar problem.

According to my assessment, the authors need to address the major and minor points listed below.

Major

1. For statistical confidence, the cross validation technique(e.g. 10-fold) should be applied for model assessment and

the results should be shown together with their confidence intervals

2. The feature set used for RF, SVM and non-convolutional ANN input doesn't seem to be adequate for the desired task.

The authors should extend the feature set and/or use entire frequency envelope data as the input.

3. The architecture of ANNs should employ widely used techniques for overfitting prevention: batch normalization/dropout. It would be

good to employ data augmentation(e.g. random shifts and crops along temporal axis) to help the CNN perform better. The batch size also

can be increased to help improve the performance and/or convergence speed.

Minor

1. SVM OVA approach requires confidence estimation. Which confidence estimation approach was used?

2. ANN architecture should be described more thoroughly: convolutional layers kernel sizes, strides, dropout/batch normalization usage,

stochastic gradient descent optimizer used, weight initialization strategy.

3. The report of Vogel et al, 2019

'Quantifying ultrasonic mouse vocalizations using acoustic analysis in a

supervised statistical machine learning framework'

solving a similar task should be mentioned

4. Strain data is not used in the analysis anyhow.

It would be interesting to compare the accuracies of two strains' shape prediction results taken separately.

Typos

96 'can not applies'

124 'postnatal (PND)'

Reviewer #2: Overview: The authors manually segmented and labeled an impressive number of mouse USVs (48699) according to categories developed by Scattoni and colleagues (2008). A sampling of 1199 USVs per category was used to train several supervised classification algorithms. Support Vector Machines (SVM), Random Forests (RF) and Artificial Neural Networks (ANN), in several configurations. The authors conclude that the best results are obtained by Support Vector Machines with the One-VS-All configuration. However, no method was particularly accurate. Precision, recall, and accuracy fell between %51.4 and 68.5% for all classifiers. I believe the moderate accuracy of the classifiers described in this report are not due to any deficiencies in methodology employed by the authors, rather they are due to the fundamental inaccuracy of human defined USV classification.

Main Issues:

The original creator of these particular USV categories, Maria Luisa Scattoni, has already published a paper using support vector machines (SVM) and random forests (RF) to categories USVs. They achieved 85% recall. What more does this paper add?

Vogel, A.P., Tsanas, A. & Scattoni, M.L. Quantifying ultrasonic mouse vocalizations using acoustic analysis in a supervised statistical machine learning framework. Sci Rep 9, 8100 (2019). https://doi.org/10.1038/s41598-019-44221-3

Experimenter derived call categories are generally falling out of favor. Substantial new evidence suggests that USVs don’t categorize neatly into discrete groups, including the data in this paper. SVMs/RFs are capable of much higher accuracy when the training data actually comes from discrete groups with clear separations. The “short” calls in the present manuscript are a well-defined group and are thus categorized accurately (>90%). But other calls like “complex” and “composite” are frequently miss-categorized. This likely isn’t a fault of the classifier. Rather, it captures the uncertainty within the human created training data.

(Discrete vs Continuous Vocalizations)

Tim Sainburg, Marvin Thielk, Timothy Q Gentner (2019) Latent space visualization, characterization, and generation of diverse vocal communication signals. bioRxiv 870311; doi: https://doi.org/10.1101/870311

Many methods are now available for un-biased call classification. These categories can then be validated through the behavioral/contextual usage of the calls. Call categories that are used identically during behavior can be collapsed. This method is more sophisticated and ethologically relevant than creating categories based on experimenter visual inspection.

Sangiamo, D.T., Warren, M.R. & Neunuebel, J.P. Ultrasonic signals associated with different types of social behavior of mice. Nat Neurosci 23, 411–422 (2020). https://doi.org/10.1038/s41593-020-0584-z

The training data, code, and finalized classifiers are the most valuable elements of this manuscript, but I did not see any indication that they will be distributed to the field. Without this, the manuscript just describes their internally used classifier with moderate accuracy.

Final Thoughts:

While there is nothing wrong with the scientific methodology employed in this paper, I would like to see all of the issues above addressed before considering the manuscript for publication.

6. PLOS authors have the option to publish the peer review history of their article (what does this mean?). If published, this will include your full peer review and any attached files.

Reviewer #1: Yes: Alexander Ivanenko

Reviewer #2: Yes: Kevin R Coffey

---

## [Author Response · Author response to Decision Letter 0]

30 Sep 2020

We would like to take this opportunity to sincerely thank the Reviewers for their valuable comments. We have carefully revised our manuscript, taking all the comments and suggestions into consideration.

General comments

First, we list the main efforts we have spent to enhance the experimental phase, in accordance with the Reviewers’ recommendations:

1) Expansion of the statistical analysis of the results;

2) Rerun of the experiments after:

a. Extending the standard feature space, for feature-based learning algorithms;

b. Improving the Convolutional Neural Network (CNN) architecture by also including pertinent techniques for overfitting prevention;

3) Implementation of two new sets of experiments, namely:

a. Classification after splitting the dataset into the two genotype-based datasets;

b. Classification performed by removing one class.

The new experiments have led to an overall improvement in the classification performance, especially when the CNN is used. Of course, the updated results are reported in the new version of the paper. 

We change the title of the manuscript in “Automatic classification of mice vocalizations using Machine Learning techniques and Convolutional Neural Networks”, suggesting the importance of use of the Convolutional Neural Networks in our paper.

Beyond the experiment aspect, another point that has importantly contributed to the paper revision is related to the work that was suggested by both Reviewers, namely:

Vogel, A.P., Tsanas, A. & Scattoni, M.L. Quantifying ultrasonic mouse vocalizations using acoustic analysis in a supervised statistical machine learning framework. Sci Rep 9, 8100 (2019).

Accordingly, in the revised paper we discuss the main points that differentiate our work from the above cited paper, which has been properly referenced therein. Furthermore, we want to point out that now we have shifted our attention to the classification method based on CNN architecture, since in the 2019 Scientific Reports paper some traditional machine learning (ML) algorithms were already tested. Consequently, the discussion on the ML techniques results to be more limited with respect to the first submission. This change of focus is also motivated by the fact that the new experiments, carried out in the revision stage, have shown notably superior performance when using CNN as stated above. In addition, as pointed out in the 3) b. item above listed, in order to make the comparison as fair as possible, we have also implemented a further set of experiments, in which the classification has been performed by removing the class harmonic, as done in Vogel’s paper.

As a final note, we also mention that we will make both the dataset and the implementation code publicly available.

In the following, we reply item by item to the specific Reviewers’ comments.

Response to Reviewer #1

Comment 1 (major)

• For statistical confidence, the cross-validation technique (e.g. 10-fold) should be applied for model assessment and the results should be shown together with their confidence intervals.

Response: Following this valuable suggestion, we have applied 10-fold cross-validation for model assessment. In the revised paper, we have reported the results as mean ± standard deviation.

Comment 2 (major) and Comment 3 (minor)

• The feature set used for RF, SVM and non-convolutional ANN input doesn’t seem to be adequate for the desired task. The authors should extend the feature set and/or use entire frequency envelope data as the input.

• The report of Vogel et al, 2019 ‘Quantifying ultrasonic mouse vocalization using acoustic analysis in a supervised statistical machine learning framework’ solving similar task should be mentioned.

Response: The problem expressed in Comment 2 (major) also implicitly deals with Comment3 (minor), therefore we would like to address both the comments in the same answer.

We agree with the Reviewer that the results obtained by using the feature set described in the original paper suggest a perfectible ability to generalize the investigated machine learning models. As a matter of fact, we have extended the original feature set by including other features extracted from the spectrograms.

The new features selection has been based from the paper suggested by the Reviewer (Voget et al., 2019). We have carefully read that report, in which the authors propose an optimal feature subset of 8 acoustic measures for USV classification. They are duration, quart 50 (start), peak freq (stddeventire), peaktopeak, max freq (stddeventire), peak freq (start), peak freq (end) and quart 75 (start). Since 4 of these 8 features are already included in the original feature set, namely, duration, peak freq (start), peak freq (end) and peak freq (stddeventire), we have just added the remaining 4 features, leading to a final feature space formed by 20 features (against the original one composed by 16 features).

The new performances outperform the ones referred to the submitted paper, even if the strongest improvement has been obtained by enhancing the CNN architecture, as pointed out in the next response.

Comment 3 (major) and Comment 2 (minor)

• The architecture of ANNs should employ widely used techniques for overfitting prevention: batch normalization/dropout. It would be good to employ data augmentation (e.g. random shifts and crops along temporal axis) to help the CNN perform better. The batch size also can be increased to help improve the performance and/or convergence speed.

• ANN architecture should be described more thoroughly: convolutional layers kernel sizes, strides, dropout/batch normalization usage, stochastic gradient descent optimizer used, weight initialization strategy.

Response: Regarding the issues detailed in Comment 3 (major), it is undoubtable that overfitting seems to afflict the proposed Convolutional Neural Network (CNN). The previously reported accuracy in training and testing phases are 87.0% and 58.3%, respectively, indicating that our model had some problem to generalize new examples not included in the training dataset.

Agreeing with the reviewer suggestions, some techniques for overfitting prevention have been implemented, listed in the following:

a. Regularization, i.e., dropout;

b. Batch normalization;

c. Testing different batch sizes;

d. Data augmentation.

In order to properly illustrate how we have implemented the just mentioned techniques, it can be helpful to start by describing the CNN architecture, that in turn also answers to Comment 2 (minor). 

First of all, we want to point out that substantial changes have been applied in the model architecture: indeed, the introduction of methods to prevent overfitting came with an increase of the architecture complexity.

The final model is composed by 5 convolutional layers constituted by 32, 64, 64, 128 and 128 filters, respectively. The convolutional layers kernel sizes are 7x7 and 5x5 in the first two layers, and 3x3 in the successive three. The stride is set to 1 for each layer.

A max-pooling layer is inserted after each convolutional layer. The pooling size is fixed to 3x3, while the stride is set to 2x2. The tensor is then flattened, and two fully connected layers are inserted, the first of size 1024 with ReLU activation function, and the second of size 10 with Softmax activation function. We mention that the Adam optimization algorithm has been used as stochastic gradient descent optimizer, and that the uniform distribution has been adopted to initialize the network weights. The description of the final CNN architecture has been thoroughly illustrated in the new version of the paper.

To prevent overfitting, the strategies that we referred above as a., b., c., and d., have been tested. Dropout, i.e. a., deletes a random sample of the activations (namely, makes them zero) during training. Of course, dropout causes information loss, in particular, losing something in the first layer propagates that loss to the whole network. We have designed several configurations, i.e., we have located dropout layers in different positions and tested a range of rates (the fraction of input units to drop), from 0.2 to 0.8. The batch normalization technique (b.) has been similarly evaluated: it helps to coordinate the update of multiple layers in the model. We have experimented batch normalization layers both before and after the activation functions, and in parallel we have considered different batch sizes (c.) in the training phase.

In the end, after this thorough hyperparameters optimization/tuning, the best model resulted the one with a dropout layer after each convolutional/max-pooling layer, except the first one, setting the rates to 0.2, 0.3, 0.4 and 0.5. Then, a further dropout layer was added after the first fully-connected layer (rate equals to 0.5), and finally batch normalization was included before the Softmax activation function of the last fully-connected layer. The batch size finally adopted is 32. 

By applying all of these adjustments, the performance of the final CNN architecture has notably improved: indeed, the test accuracy has been enhanced from 58.3% to 78.8%.

Here is a final note on data augmentation (d.): given the characteristics of the spectrograms, some image transformations cannot be analyzed, since they would invalidate the ground-truth classification. For example, rotation or horizontal/vertical flipping could lead the network to confuse the upward class with the downward class, and vice-versa. For this reason, and also as suggested by the reviewer, we have increased the dataset by shifting and cropping the spectrogram images at random: however, the experiments have shown that data augmentation does not significantly improve the performance of the model. As with the other points, the discussion on overfitting prevention has been properly argued in the revised paper.

Comment 1 (minor)

• SVM OVA approach requires confidence estimation. Which confidence estimation approach was used? 

Response: For SVM OVA method, we set a maximum confidence strategy, similar to the weighted voting strategy from OVO systems. So, the output class is taken from the classifier with the largest positive answer. We have reported this detail in the revised version of the paper.

Comment 4 (minor)

• Strain data is not used in the analysis anyhow. It would be interesting to compare the accuracies of two strains' shape prediction results taken separately. 

Response: We have taken into consideration this valuable suggestion. As a matter of fact, in the revised paper we have included a new set of experiments, by testing the classification methods on the genotype-based datasets taken separately.

Typos

• 96 'can not applies' 

• 124 'postnatal (PND)' 

Response: We have corrected the typos indicated by the Reviewer.

Response to Reviewer #2

Comment 1

• The original creator of these particular USV categories, Maria Luisa Scattoni, has already published a paper using support vector machines (SVM) and random forests (RF) to categories USVs. They achieved 85% recall. What more does this paper add?

Vogel, A.P., Tsanas, A. & Scattoni, M.L. Quantifying ultrasonic mouse vocalizations using acoustic analysis in a supervised statistical machine learning framework. Sci Rep 9, 8100 (2019). 

Response: We have meticulously read the suggested report, and we agree that it is important to shed light on the improvements led by our work with respect to Vogel’s one. We want to focus here on the two main breakthroughs that in our opinion are worthy to be pointed out.

First, we made a significant effort to build our dataset. We performed our experiments on 48699 samples, which is a much more sizable dataset than the 225 samples evaluated in the suggested paper. It is well-known that the bigger the sample size is, the more accurate the research results are. That is because larger dataset cardinality allows to better determine the average values of data and reduce potential errors from testing a small number of possible outliers. As a consequence, the statistical analysis becomes more accurate and the margin of error smaller. Given these crucial considerations, we believe that our results benefit from a very effective statistical soundness, and thus that they may represent a further valuable benchmark for future works in this research field. Of course, the dataset will be made publicly available.

The second aspect we want to mention is that in our work we explored the classification ability of a Convolutional Neural Network (CNN), extending the analysis beyond traditional machine learning algorithms, such as Support Vector Machine and Random Forest, employed in Vogel’s paper. To the best of the authors’ knowledge, this work is the first to propose a deep learning architecture aimed to classify the ten categories originally introduced in [17]. Furthermore, the CNN model has been notably modified during the revision phase. Accordingly, we have included all of these modifications in the revised paper. In particular, if compared with the original proposed architecture, we have changed the network topology by inserting more convolutional layers and increased the number of the filters composing them, in parallel to have investigated some techniques for overfitting prevention, such as regularization (drop out) and batch normalization (please find more details in the new version of the paper). As a consequence of such improvements, the performance of the final CNN model now achieves an accuracy of 78% on the test dataset, strongly improving the results reported in the original paper (where the test accuracy was 58.3 %).

• Experimenter derived call categories are generally falling out of favor. Substantial new evidence suggests that USVs don’t categorize neatly into discrete groups, including the data in this paper. SVMs/RFs are capable of much higher accuracy when the training data actually comes from discrete groups with clear separations. The “short” calls in the present manuscript are a well-defined group and are thus categorized accurately (>90%). But other calls like “complex” and “composite” are frequently miss-categorized. This likely isn’t a fault of the classifier. Rather, it captures the uncertainty within the human created training data.

(Discrete vs Continuous Vocalizations)

Tim Sainburg, Marvin Thielk, Timothy Q Gentner (2019) Latent space visualization, characterization, and generation of diverse vocal communication signals. bioRxiv 870311.

Response: We think that this different perspective is very interesting. We agree that automatic USVs classification into a fixed number of discrete groups is a challenging task, as also suggested by the paper cited in the comment. Indeed, the results reported in the originally submitted paper confirmed the difficulty of distinguishing particular classes, such as complex and composite.

Nevertheless, even if manual classification is an undeniably time-consuming activity and also vulnerable to personal interpretation, it can be performed within a more than reasonable margin of error, especially when the general shape of the spectrogram is analyzed.

Even acknowledging an intrinsic uncertainty in the human labelling of data, in our work we addressed the problem of designing efficient and accurate machine learning and deep learning models, aimed to automatically classify vocalizations starting from the audio tracks. The performance obtained from the new set of experiments suggests that crucial improvements have been implemented to the originally proposed models. Indeed, the new results show a lesser uncertainty in USVs categorization, indicating that well defined models can actually lead to significant performance even when discrete vocalizations classification is investigated.

This discussion has been inserted also in the revised paper, citing the suggested article.

Comment 2

• Many methods are now available for un-biased call classification. These categories can then be validated through the behavioral/contextual usage of the calls. Call categories that are used identically during behavior can be collapsed. This method is more sophisticated and ethologically relevant than creating categories based on experimenter visual inspection.

Sangiamo, D.T., Warren, M.R. & Neunuebel, J.P. Ultrasonic signals associated with different types of social behavior of mice. Nat Neurosci 23, 411–422 (2020). 

Response: In the literature different methods of calls classification exist. We used a method based on experimenter visual inspection referring to papers of Dr. Scattoni cited in our references. This method permits to the operator to classify vocalizations based on features as frequency content, duration, amplitude and general shape. This classification provides a detailed USV characterization, but it says nothing about the meaning of the calls. Recently, new technologies have been developed such as a microphone array system and sound source localization method, to localize and assign USVs to individual mice during a social context (Heckman et al., 2017 doi:10.1038/s41598-017-02954-z; Warren et al., 2018 doi:10.1016/j.jneumeth.2017.12.013; Sangiamo et al., 2020 doi.org/10.1038/s41593-020-0584-z). This permits to associate distinct vocalizations categories with different types of murine social behavior giving important information to calls meaning. Unfortunately, we do not have these sophisticated instruments to associate behaviors to calls. In addition, we recorded vocalizations emitted by pups and not adults and in pups it is not possible to associate calls to behaviors (walk, fight, mutual circle… in a social context) as done by Sangiamo and colleagues because pups are still very young (first days after birth) and unable to move or perform more complex actions. They still have closed eyes. Finally, we agree with the Reviewer with the idea that it is very interesting to understand the meaning of mice calls and so, in the future, we would like to perform more sophisticated and ethologically relevant analysis on adult mice.

Comment 3

• The training data, code, and finalized classifiers are the most valuable elements of this manuscript, but I did not see any indication that they will be distributed to the field. Without this, the manuscript just describes their internally used classifier with moderate accuracy.

Response: We are in total agreement with the Reviewer. As a matter of fact, we will make both the dataset and the implementation code publicly available.

---

## [Decision Letter · Decision Letter 1]

6 Nov 2020

PONE-D-20-09612R1

Automatic classification of mice vocalizations using Machine Learning techniques and Convolutional Neural Networks

PLOS ONE

Dear Dr. Premoli,

Thank you for submitting your manuscript to PLOS ONE. After careful consideration, we feel that it has merit but does not fully meet PLOS ONE’s publication criteria as it currently stands. Therefore, we invite you to submit a revised version of the manuscript that addresses the points raised during the review process.

Please, revise the manuscript to address the concerns raised by the reviewer including changes of the presentation for clarity, providing omitted results, data analysis justification and appropriately citing relevant studies.

1. The results of all classifiers mentioned in the paper should be presented in a table.

If the resulting accuracy is especially bad and is not worth considering than it should be stated explicitly.

2. Please, clarify and directly state which approach for data augmentation was used for the best-performing CNN.

Minor

Abstract

1. tested some supervised ...  -> tested some other supervised ...

2. extracted by the spectrograms -> extracted from the spectrograms

Introduction

1. "... on such a fixed repertoire of calls typologies".  May be "on a predefined set of call types" ?

USV Recording and analysis

1. "postnatal (PND) 6, 8 ..." -> "postnatal day (PND) 6, 8..."

2. Threshold and hold time parameter values used in Avisoft for vocalization extraction should be provided here.

Description of the experiments

1. 'the first step consisted into' -> 'the first step consisted of'

2. 'giving this way a statistical soundness' -> 'ensuring the statistical soundness' ?

Features extraction section

1. 'identified by the color...'  -> 'identified by the brightness...'

Support vector machines

1. 'Maximum confidence strategy was used...' - how the confidence was estimated? SVM provides binary result out of the box, the actual confidence assessment approaches vary.

We look forward to receiving your revised manuscript.

Kind regards,

Gennady Cymbalyuk, Ph.D.

Academic Editor

PLOS ONE

Reviewers' comments:

Reviewer's Responses to Questions

**Comments to the Author**

1. If the authors have adequately addressed your comments raised in a previous round of review and you feel that this manuscript is now acceptable for publication, you may indicate that here to bypass the “Comments to the Author” section, enter your conflict of interest statement in the “Confidential to Editor” section, and submit your "Accept" recommendation.

Reviewer #1: All comments have been addressed

Reviewer #2: All comments have been addressed

2. Is the manuscript technically sound, and do the data support the conclusions?

Reviewer #1: Yes

Reviewer #2: Yes

3. Has the statistical analysis been performed appropriately and rigorously? 

Reviewer #1: Yes

Reviewer #2: Yes

4. Have the authors made all data underlying the findings in their manuscript fully available?

Reviewer #1: No

Reviewer #2: Yes

5. Is the manuscript presented in an intelligible fashion and written in standard English?

Reviewer #1: No

Reviewer #2: Yes

6. Review Comments to the Author

Reviewer #1: Though the authors addressed the points noted in the previous review, the document undergone major

changes that gave rise to several more major and minor inconsistencies to be addressed.

Additionally, for the next review round, I would ask the authors to add line numbers in the document

and to provide the source code for the models described in the paper.

Major

1. The Random Forests, MP, Stacking NN classification performance results are completely omitted in the paper.

SVM performance results are hard to find in the text. At the same time, both RF and SVM are mentioned in the abstract and

multiple times throughout the text. The results of all classifiers mentioned in the paper should be presented in a table.

If the resulting accuracy is especially bad and is not worth considering than it should be stated explicitly.

Note that the results are compared with the results of RF classifier in Vogel et al, 2019 paper, so the omitting of the RF results

looks strange.

2. 'Data-preprocessing' paragraph. 'This operation prevent imbalanced dataset...' - usually, that is not the main

reason to apply data augmentation when training CNN. To deal with imbalanced data one can just use weighted loss function, as

you mention further in the paragraph. Data augmentation is used mostly to prevent overfitting,

generating a potentially infinite, though maybe not diverse enough set of training samples.

From your description it is not clear which approach for data augmentation

is used for the best-performing CNN: statically generated examples with random crops and shifts,

just to balance the source data set,

or the on-the-fly generation of randomly altered samples during the training,

together/without using loss function weighting?

That should be stated explicitly.

Additionally, I think that is not necessary to explain that one should not rotate on flip images in that case.

'data augmentation' is a general term for generation of pseudo-diverse samples and

that is quite obvious that one should not use the techniques applied for object photographs classification here...

3. 'Proposed classification methods' paragraph : you briefly explain the principles of CNN and only after that you start

explaining trivial theory of MP, talking about weights, activation functions and neurons.

'Multilayer perceptron' sub-section should go first, since it describes the most basic things on which CNN is based as well.

Furthermore, I don't think Fig.2 is worth placing in the paper. That diagram is quite trivial, it occupies half of a page

and is known since 70s. It can be found in any textbook about ANN basics.

Why not to use that space to depict the actual architecture of the CNN + ANNs you designed,

maybe together with some trained kernel weights visualization or other data?

For example, see the figures in our recently published work

Ivanenko et al, 2020, "Classifying sex and strain from mouse ultrasonic vocalizations using deep learning".

Minor

Abstract

1. tested some supervised ... -> tested some other supervised ...

2. extracted by the spectrograms -> extracted from the spectrograms

Introduction

1. "... on such a fixed repertoire of calls typologies". I think this should be rephrased.

May be "on a predefined set of call types" ?

2. Please cite our paper

Ivanenko et al, 2020, "Classifying sex and strain from mouse ultrasonic vocalizations using deep learning", PLOS CB

in the introduction. Though we don't use Scattoni classical vocalization types there, we also

classify vocalizations using CNNs basing on their spectrogram shape (ascending, descending, number of jumps, peaks,

complexity etc), thus implementing the 'top-down' approach you mentioned.

5. 'Simulation results' - I think the word 'simulation' is misleading here.

USV Recording and analysis

1. "postnatal (PND) 6, 8 ..." -> "postnatal day (PND) 6, 8..."

2. Threshold and hold time parameter values used in Avisoft for vocalization extraction should be provided here.

Description of the experiments

1. 'the first step consisted into' -> 'the first step consisted of'

2. 'giving this way a statistical soundness' -> 'ensuring the statistical soundness' ?

Features extraction section

1. 'identified by the color...' -> 'identified by the brightness...'

Support vector machines

1. 'Maximum confidence strategy was used...' - how the confidence was estimated? SVM provides binary result out of the box,

the actual confidence assessment approaches vary.

Reviewer #2: The authors have made significant efforts to improve their CNN based classification architecture and have at least discussed and considered the theoretical limitations I posed in review. I agree that the scale of the new experiment and improved classification accuracy now expand upon, rather then duplicate, the work of the Scattoni Lab. With the addition of a publicly available dateset and classification CNN, this work now makes a tangible contribution to the field and I recommend it be accepted for publication.

7. PLOS authors have the option to publish the peer review history of their article (what does this mean?). If published, this will include your full peer review and any attached files.

Reviewer #1: **Yes: **Alexander Ivanenko

Reviewer #2: **Yes: **Kevin R Coffey

---

## [Author Response · Author response to Decision Letter 1]

4 Dec 2020

We would like to take this opportunity to sincerely thank everyone involved in the review process. We are glad that the adjustments introduced in response to reviewers' comments process have been generally appreciated. The further changes suggested by the reviewer #1 are detailed below.

Response to Reviewer #1

Comment 1 (major)

 The Random Forests, MP, Stacking NN classification performance results are completely omitted in the paper. SVM performance results are hard to find in the text. At the same time, both RF and SVM are mentioned in the abstract and multiple times throughout the text. The results of all classifiers mentioned in the paper should be presented in a table. If the resulting accuracy is especially bad and is not worth considering than it should be stated explicitly.

Note that the results are compared with the results of RF classifier in Vogel et al, 2019 paper, so the omitting of the RF results looks strange.

Response: As mentioned in the previous response process, in the first revised paper we had moved the focus from the “numerical features”-based classification techniques, both for distancing our work with respect to Vogel’s paper (as requested by both the reviewers), and also because the new results obtained by using the Convolutional Neural Networks had outperformed the others. Accordingly, we had considered not interesting and redundant the list of all the performance for all the investigated techniques, with that not for that intending to diminish the importance of studying those methods in our work.

However, we agree with the reviewer that a table containing the overview of all the results may benefit the completeness of the paper. As a matter of fact, in the new version of the paper, we have inserted such a table in the “Performance analysis” section. Nonetheless, to streamline the paper, we just report the performance related to the entire mice vocalizations dataset, since the strain-based performances result are similar to the ones obtained for the entire dataset, and so they do not add significant extra information.

Finally, the source code has been properly integrated as well.

Comment 2 (major)

 'Data-preprocessing' paragraph. 'This operation prevents imbalanced dataset...' - usually, that is not the main reason to apply data augmentation when training CNN. To deal with imbalanced data one can just use weighted loss function, as you mention further in the paragraph. Data augmentation is used mostly to prevent overfitting,

generating a potentially infinite, though maybe not diverse enough set of training samples.

From your description it is not clear which approach for data augmentation

is used for the best-performing CNN: statically generated examples with random crops and shifts, just to balance the source data set, or the on-the-fly generation of randomly altered samples during the training, together/without using loss function weighting? That should be stated explicitly.

Additionally, I think that is not necessary to explain that one should not rotate on flip images in that case. 'data augmentation' is a general term for generation of pseudo-diverse samples and that is quite obvious that one should not use the techniques applied for object photographs classification here...

Response: Thanks for pointing this out. We take this space to better clarify how data augmentation has been used in our experiments, and we will report these considerations in the new version of the paper.

For the CNN design, we have tested data augmentation for both overfitting and imbalanced dataset prevention, separately. In the first case, an offline generation of new examples with random shifts and crops is applied over the entire training dataset; when using data augmentation for this aim, a properly weighted loss function is employed during the training in order to handle the imbalanced dataset problem too. In the second one, data augmentation is performed to balance the source dataset, and so by generating new samples just for the under-represented classes.

However, note that data augmentation has been tested in the experimental phase, but it has not been in the end used to obtain the final best-performing CNN model. Indeed, the experimental tests have revealed that just including dropout and batch normalization layers is the most efficient action for overfitting prevention (as reported in the “Classification via CNN” section). Furthermore, the experiments have also provided more uniform accuracy values across the classes when the loss function is properly weighted instead of repopulating the under-represented classes by implementing data augmentation. In conclusion, data augmentation has not been exploited for designing the most-performing CNN architecture.

As mentioned before, in the new version of the paper, we have better clarified all these aspects by partially re-writing the former section “Data pre-processing”, which is now titled “Class imbalance handling”. Therein, the reviewer will find that all his/her observations above have been properly considered. 

Comment 3 (major)

 'Proposed classification methods' paragraph: you briefly explain the principles of CNN and only after that you start explaining trivial theory of MP, talking about weights, activation functions and neurons. 'Multilayer perceptron' sub-section should go first, since it describes the most basic things on which CNN is based as well.

Furthermore, I don't think Fig.2 is worth placing in the paper. That diagram is quite trivial, it occupies half of a page and is known since 70s. It can be found in any textbook about ANN basics.

Why not to use that space to depict the actual architecture of the CNN + ANNs you designed, maybe together with some trained kernel weights visualization or other data? For example, see the figures in our recently published work

Ivanenko et al, 2020, "Classifying sex and strain from mouse ultrasonic vocalizations using deep learning".

Response: Thanks for the suggestion. We agree with the reviewer that the paper may be clearer by swapping the “Convolutional Neural Networks” and “Multilayer Perceptron” sub-sections.

In the revised version of the paper, we have modified the description of the multilayer perceptrons without the support of Fig. 2. In its place, there is a new figure depicting the entire architecture of the designed CNN; such a figure has been suitably added in “Classification via CNN” section.

Minor (Abstract)

 tested some supervised ... -> tested some other supervised ...

 extracted by the spectrograms -> extracted from the spectrograms

Response: We have revised both the sentences as in 1) and 2).

Minor (Introduction)

 "... on such a fixed repertoire of calls typologies". I think this should be rephrased. May be "on a predefined set of call types"?

 Please cite our paper Ivanenko et al, 2020, "Classifying sex and strain from mouse ultrasonic vocalizations using deep learning", PLOS CB in the introduction. Though we don't use Scattoni classical vocalization types there, we also classify vocalizations using CNNs basing on their spectrogram shape (ascending, descending, number of jumps, peaks, complexity etc), thus implementing the 'top-down' approach you mentioned.

 'Simulation results' - I think the word 'simulation' is misleading here.

Response: We have revised both the sentences as in 1) and 3). We have also carefully read the paper suggested in 2) and cited it in the introduction while mentioning the top-down approach.

Minor (USV Recording and analysis)

 "postnatal (PND) 6, 8 ..." -> "postnatal day (PND) 6, 8..."

 Threshold and hold time parameter values used in Avisoft for vocalization extraction should be provided here.

Response: We have revised the sentences as in 1) and inserted the requested parameters in the appropriate section as suggested in 2).

Minor (Description of the experiments)

 'the first step consisted into' -> 'the first step consisted of'

 'giving this way a statistical soundness' -> 'ensuring the statistical soundness'?

Response: We have revised both the sentences as in 1) and 2).

Minor (Feature extraction section)

 identified by the color...' -> 'identified by the brightness...'

Response: We have revised the sentences as in 1).

Minor (Support Vector Machines)

 'Maximum confidence strategy was used...' - how the confidence was estimated? SVM provides binary result out of the box, the actual confidence assessment approaches vary.

Response: The confidence is directly given by the decision function, which is the signed distance between the tested sample and the separating hyperplane. More precisely, the decision function is computed as ∑_(i∈SV)▒〖y_i α_i K(x_i,x)+b〗, where

 x is the sample to predict;

 x_i are the support vectors (SV) that construct the hyperplane;

 b is the intercept;

 α_i are the dual coefficients computed in the SVM optimization problem;

 y_i are the binary labels corresponding to x_i;

 K(x_i,x)=φ(x_i )^T φ(x) is the employed kernel.

In the OVA-strategy, we generate 10 classifiers: the first separates class 1 from the remaining ones, the second does the same for class 2, and so on for all 10 classes. The decision function is computed for each class as reported in the previous equation. The classifier that returns the largest positive value identifies the winner class.

---

## [Decision Letter · Decision Letter 2]

15 Dec 2020

Automatic classification of mice vocalizations using Machine Learning techniques and Convolutional Neural Networks

PONE-D-20-09612R2

Dear Dr. Premoli,

We’re pleased to inform you that your manuscript has been judged scientifically suitable for publication and will be formally accepted for publication once it meets all outstanding technical requirements.

Please, consider minor edits suggested by one of the reviewers.

Kind regards,

Gennady Cymbalyuk, Ph.D.

Academic Editor

PLOS ONE

Additional Editor Comments (optional):

Reviewers' comments:

Reviewer's Responses to Questions

**Comments to the Author**

1. If the authors have adequately addressed your comments raised in a previous round of review and you feel that this manuscript is now acceptable for publication, you may indicate that here to bypass the “Comments to the Author” section, enter your conflict of interest statement in the “Confidential to Editor” section, and submit your "Accept" recommendation.

Reviewer #1: All comments have been addressed

Reviewer #2: All comments have been addressed

2. Is the manuscript technically sound, and do the data support the conclusions?

Reviewer #1: Yes

Reviewer #2: Yes

3. Has the statistical analysis been performed appropriately and rigorously? 

Reviewer #1: Yes

Reviewer #2: Yes

4. Have the authors made all data underlying the findings in their manuscript fully available?

Reviewer #1: Yes

Reviewer #2: Yes

5. Is the manuscript presented in an intelligible fashion and written in standard English?

Reviewer #1: Yes

Reviewer #2: Yes

6. Review Comments to the Author

Reviewer #1: The authors carefully addressed all the comments given in the previous review round. I reccommend acceptance of the paper.

Some minor grammar/stylistic/logic errors could be considered:

line 391 : generating to -> generating of/for

line 432: the CNN architecture has outperformed the other standard features-based methods ->

the CNN architecture has outperformed the standard features-based methods (CNN is not a feature-based method in the paper ...)

line 455: are arranged on the ten columns -> are arranged in ten columns

line 458 : The values into the matrices are normalized -> the values in the matrices ....

line 463, 502: on the x axis there are the predicted labels : "on x axis there are..." sounds a bit ungrammatical to me; usually they use something like "x axis refers to ... " to describe the meaning of axes

line 519: The performance showed that by exploiting the whole time/frequency information of the spectrogram

leads to significantly higher performance than considering a subset

->

The performance showed that the exploiting of the whole time/frequency information of the spectrogram

leads to significantly higher performance than considering only a subset of numerical features

line 525: The final set up on an automatic classification method will

definitely solve the current main problems in USVs manual classification: long time consuming and

operator-dependent.

->

The final set up of(?) an automatic classification method will

definitely solve the current main problems of USVs manual classification: its being a time consuming process and

operator bias.

Reviewer #2: The authors have done a good job responding to the additional reviewers comments. I recommended to accept the paper in the previous revision.

7. PLOS authors have the option to publish the peer review history of their article (what does this mean?). If published, this will include your full peer review and any attached files.

Reviewer #1: **Yes: **Aleksandr Ivanenko

Reviewer #2: **Yes: **Kevin Coffey

---

## [Editor Report · Acceptance letter]

7 Jan 2021

PONE-D-20-09612R2 

Automatic classification of mice vocalizations using Machine Learning techniques and Convolutional Neural Networks 

Dear Dr. Premoli:

I'm pleased to inform you that your manuscript has been deemed suitable for publication in PLOS ONE. Congratulations! Your manuscript is now with our production department. 

Kind regards, 

on behalf of

Dr. Gennady Cymbalyuk 

Academic Editor

PLOS ONE